# How scent and nectar influence floral antagonists and mutualists

**Danny Kessler\*, Mario Kallenbach, Celia Diezel, Eva Rothe, Mark Murdock[†], Ian T Baldwin\***

Department of Molecular Ecology, Max-Planck Institute for Chemical Ecology, Jena, Germany

**Abstract** Many plants attract and reward pollinators with floral scents and nectar, respectively, but these traits can also incur fitness costs as they also attract herbivores. This dilemma, common to most flowering plants, could be solved by not producing nectar and/or scent, thereby cheating pollinators. Both nectar and scent are highly variable in native populations of coyote tobacco, *Nicotiana attenuata*, with some producing no nectar at all, uncorrelated with the tobacco's main floral attractant, benzylacetone. By silencing benzylacetone biosynthesis and nectar production in all combinations by RNAi, we experimentally uncouple these floral rewards/attractants and measure their costs/benefits in the plant's native habitat and experimental tents. Both scent and nectar increase outcrossing rates for three, separately tested, pollinators and both traits increase oviposition by a hawkmoth herbivore, with nectar being more influential than scent. These results underscore that it makes little sense to study floral traits as if they only mediated pollination services.

**\*For correspondence:** dkessler@ice.mpg.de (DK); baldwin@ice.mpg.de (ITB)

**Present address:** [†]University of Pittsburgh, Pennsylvania, United States

## Introduction

Plants attract pollinators to move pollen from one flower to another to ensure outcrossing and frequently pay for these pollination services by rewarding floral visitors with nectar (*Kevan and Baker, 1983*; *Schemske and Bradshaw, 1999*; *Raguso and Willis, 2005*). Even though it is clear that both floral scent and nectar provide fitness benefits for plants, rewardless flowers have evolved in all major groups of angiosperms (*Renner, 2006*). Orchids, in particular, have frequently evolved deceptive pollination systems, in which flowers attract pollinators by mimicking mating partners or oviposition sites without offering rewards (*Schiestl, 2005*). But rewardless, nectar-free, flowers are commonly found within species that normally provide nectar, and this is surprising, as the occasional nectar-free flowers would have a disadvantage if visitors have the sensory abilities to avoid rewardless flowers (*Karban, 2015*). Early on, theorists (*Bell, 1986*) recognized that most flowers hide their rewards, for example, deep in the corolla tube, which thwarts the easy visual evaluation of a flower's standing nectar volume and developed an ESS model for the proportion of cheating flowers and discriminating visitors that would be evolutionarily stable. Researchers have since uncovered evidence consistent with the predictions of the model. *Gilbert et al. (1991)* found that nectar secretion was highly variable within plants of a population and suggested that floral visitors could distinguish between low and high nectar secreting plants. Recent research suggests that hawkmoth pollinators can use humidity as a proxy for the presence of nectar (*von Arx et al., 2012*).

To examine the importance of nectar for pollination services and to study the fitness advantages of nectar-cheating plants, researchers have used native varieties with reduced nectar accumulation, introgression lines (*Brandenburg et al., 2012a*, *2012b*), artificial flowers (e.g., *Ishii et al., 2008*), or conducted direct manipulations of nectar quantities by adding artificial nectar to flowers (e.g., *Mitchell and Waser, 1992*; *Jersáková et al., 2008*). *Ishii et al. (2008)* found that pollinators avoided inflorescences with greater numbers of empty flowers. *Smithson (2002)* added nectar to

**eLife digest** Flowering plants have evolved a number of different approaches to reproduction. Some use their own pollen and self-fertilize, while others use pollen from other nearby plants. This fertilization by other plants is called 'outcrossing' and introduces new genetic variation into each generation, which is extremely important for the evolutionary process.

Some flowering plants rely on animals to help with pollination, attracting visitors with floral scents and rewarding the visitors with sugar-rich nectar. But scent and nectar also attract herbivores that damage the plants. This causes a dilemma for flowering plants, which has led some to evolve to not produce scent or to offer no nectar while masquerading as a plant that does. Previous studies into the costs and benefits of such strategies have looked at the effects of either floral scent or nectar, but no-one has uncoupled the effects of these two traits on both pollination and herbivore attack.

Kessler et al. have addressed this issue in wild tobacco plants, which can both self-fertilize and outcross, and which produce varying amounts of scent and nectar. The experiments were conducted under mesh tents and in field trials in the plant's natural habitat: the Great Basin Desert in Utah. Kessler et al. used a gene-silencing technique called 'RNA interference' to inhibit the production of scent or nectar, either separately or together. When grown in field trials, under conditions that prevent self-fertilization, these tobacco plants are frequently visited by a hummingbird and three species of hawkmoth. All four of these animals pollinate the tobacco plants, but one of the moths also lays eggs that hatch into caterpillars, which damage the plant. Kessler et al. monitored the effects that the loss of scent, nectar or both had on visits by each pollinator and on outcrossing.

These experiments revealed that scent is essential to attract one hawkmoth species but not for another (called *Hyles lineata*). Furthermore, while, the hummingbird needs nectar, the *H. lineata* moth does not; but this moth won't visit flowers that lack both scent and nectar. The experiments also show that, for the moth that lays its eggs on the tobacco plants, both scent and nectar increase pollination and egg laying, but nectar has a stronger effect. Thus reducing nectar, as this tobacco plant does in the wild, is one strategy that can be used to reduce herbivore attack by caterpillars. Together, these findings highlight that it is important to study both herbivores and pollinators when attempting to understand the evolution of floral traits.

rewardless orchids, and while the addition changed bee behavior, it did not influence plant fitness. *Brandenburg et al. (2012a)* found that nectar-deficient *Petunia* lines produced fewer seeds than did nectar-replete control plants, because *Manduca sexta* moths reduced their probing times in low-nectar plants, which in turn, reduced pollen transfer and thus seed set.

While rewards keep pollinators moving pollen from one plant to another, other cues, such as floral scent, provide honest signals that advertise the occurrence of the rewards (*Wright and Schiestl, 2009*). Floral scent is known to play a central role in attracting insect pollinators to flowers (*Galen and Newport, 1988*; *Jürgens et al., 2002*; *Klahre et al., 2011*; *Byers et al., 2014*; *Riffell et al., 2014*). The effect of floral scent on the pollination success of single pollinator species has been studied mainly with scent augmentations and additions to existing scent bouquets (e.g., *Majetic et al., 2009*; *Shuttleworth and Johnson, 2010*). *Shuttleworth and Johnson (2010)*, for example, showed that single sulphur compounds are responsible for the shift between wasp and fly pollination in *Eucomis* (Hyacinthaceae). *Byers et al. (2014)* found altered bumblebee visitation rates in response to single volatile compounds which were added to the scent bouquet of *Mimulus* species. In most studies, only one pollinator species was investigated at a time, frequently in very specialized model systems, often the sexually deceptive pollination systems of orchids (*Schiestl, 2005*; *Schiestl and Schlüter, 2009*). Several studies investigated fitness outcomes of these manipulations. *Majetic et al. (2009)* for example, found a positive effect on both pollinator visitation and seed production in *Hesperis matronalis* by augmenting inflorescences with scent extracts. *Kessler et al. (2008)* genetically manipulated the biosynthesis of the most abundant floral volatile, benzylacetone (BA), in the flowers of the wild tobacco (*Nicotiana attenuata*) to demonstrate that BA emission is required to maximize both maternal and paternal fitness in the field when the entire native community of floral visitors had access to the flowers. Few studies have investigated the influence of scent and nectar on the entire pollinating community simultaneously.

Floral traits are shaped not only by interactions with mutualists, but also with antagonist, such as herbivores, florivores, seed feeders, or nectar robbers (*Strauss and Whittall, 2006*; *Andrews et al., 2007*; *Kessler et al., 2013*). Both floral scent and nectar are known to influence antagonists and again manipulative experiments have been essential in illuminating these interactions. Floral scents allow florivores to locate their host plants (*Theis, 2006*; *Theis and Adler, 2012*). Several studies have shown that individual compounds in floral scent bouquets specifically deter florivores and nectar robbers (*Galen et al., 2011*; *Junker et al., 2011*; *Kessler et al., 2013*) and thus complex floral scents can both attract pollinators and deter antagonists. Nectar quantity has also been shown to influence herbivory, as inferred by adding nectar or sugar solutions to flowers to increase the standing volume of nectar. In both *Datura wrightii* (*Adler and Bronstein, 2004*), as well as *N. attenuata* (*Kessler, 2012*) nectar addition increased oviposition by *M. sexta*, a pollinator as an adult as well as a devastating herbivore as a caterpillar for both plant species. The inference from both studies is that plants could reduce their herbivore load by producing only small amounts or no nectar at all.

As suggested by the early ESS modeling efforts (*Bell, 1986*), a few nectar-free plants dispersed in a large population of nectar-producing plants could realize a fitness benefit from reduced herbivore loads, if at low frequencies pollinators cannot learn their locations (*Gumbert and Kunze, 2001*) or otherwise identify their lack of rewards (*Smithson and MacNair, 1997*; *Smithson, 2009*). A few studies investigated the influence of floral traits for both pollinators and herbivores at the same time. *Theis et al. (2014)*, for example, found that floral sesquiterpenoids from Cucurbitaceae were the best predictors of flower preference for both the specialist pollinator squash bees and the specialist herbivore cucumber beetle. *Schiestl et al. (2014)* showed that the herbivory-induced reduction of floral volatiles reduced the attractiveness of flowers to pollinators. These studies however were focusing on floral signals and did not consider the function of floral rewards, and so while the ESS explanation for rewardless flowers being maintained by frequency dependent selection is well established (*Bell, 1986*), we still lack studies that evaluate the relative selective pressures of mutualistic and antagonistic floral visitors on floral reward provisioning strategies.

In this study, we rigorously investigate the consequences of reducing floral scent and nectar independently and simultaneously by RNAi not only for mutualistic interactions but also for oviposition rate, which is an excellent predictor of future herbivory. We measured outcrossing rates afforded by the entire natural pollinator community in *N. attenuata's* native habitat, by three specific pollinators (hummingbirds and two hawkmoth species), and oviposition by *M. sexta*. Different approaches were used in the field and laboratory to study pollination and herbivory. At our field site at the Lytle Ranch Preserve in SW Utah, we conducted common garden experiments using plants silenced by RNAi in the production of floral scent (CHAL; *Kessler et al., 2008*), floral nectar (SWEET9; *Lin et al., 2014*), or both (CHALxSWEET9), in comparison to empty vector-transformed control plants (EV). The RNAi constructs silenced only the targeted pathways and were otherwise isogenic, and the transformed plants were morphologically indistinguishable from EV transformed and wild-type (WT) plants. This approach allowed for the study of the entire panoply of interactions simultaneously, including different floral visitors, and herbivores to gain an unbiased picture of the interactions of individual players in the complex web of interactions that occur when flowers advertise for pollinator services.

## Results and discussion

To examine the natural genetic variability in floral advertisement and rewards for pollinator services in native populations of *N. attenuata* plants, seeds from plants from 13 native populations, collected between 1993 and 2009 within a 200 km radius of our field station in the SW USA (Utah, Arizona), were used. The variance in standing nectar volume was assessed by selecting 52 plants in the glasshouse. The average standing nectar volume was 3.3 µL and varied between single genotypes between 1.3 µL and 5.7 µL per flower (*Figure 1*). In addition to this fourfold difference in nectar volume, we also assessed the variability of BA emission from individual flowers from the same sample of plant populations. The average emission was 5.0 ng BA per flower per night, varying between a minimum emission of 0.5 ng and a maximum of 33.9 ng, equivalent to a 70-fold difference in emission rates (*Figure 1*). Nectar sugar concentrations were in contrast, rather constant. Sugar concentration in newly opened flowers varied between 14.6 and 22.8%, with an average of 18.2%.

In a second screening, 424 individual plants were screened for the presence or absence of nectar and scent. Seeds for this experiment were collected from 75 natural populations in Utah, Arizona, and

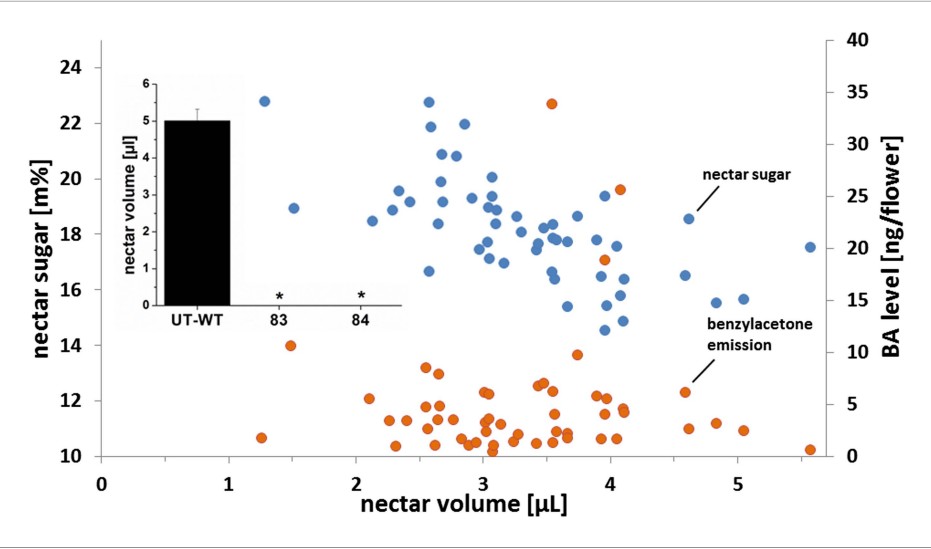

**Figure 1**. Characterization of 52 plants from different natural populations. Mean standing nectar volume (n = 6 to 8 flowers/plant), nectar sugar concentration (n = 6 to 8 flowers/plant), and benzylacetone (BA) level (n = 1 flower/plant). Nectar volume and sugar concentration were measured from newly opened flowers at the end of the nectar production period (5–6 am). BA level was measured over the night (8 pm–6 am) and calculated from peak areas normalized using tetralin as an internal standard. Inset: mean (+SE) standing nectar volume in newly opened flowers at the end of the nectar production period (5–6 am) in three native phenotypes (Ut-WT, 83, and 84; n = 7–9) found within a separate sample of 424 native *Nicotiana attenuata* plants that were screened for the presence of nectar. Asterisks indicate significant differences, as informed by a nonparametric Mann–Whitney U-test p < 0.05.

California between 1990 and 2009. Of these, two plants produced no nectar at all. These lines (83 and 84) were inbred for two generations and plants of the $T_2$ generations of both lines also did not produce any nectar in any flowers at all stages of development (*Figure 1*), demonstrating that the lack of nectar production was heritable. All flowers from all 424 plants in this analysis produced BA in some measureable quantity. We found no evidence for a linear correlation between BA emission and nectar volume (y = 0.1875x + 1.02, $R^2$ = 0.01), BA emission and nectar sugar concentration (y = 0.1233x + 3.89, $R^2$ = 0.01), or nectar volume and nectar sugar (y = −1.4471x + 22.99, $R^2$ = 0.41). While we cannot exclude that genotypes completely lacking floral scent occur in native populations, our results demonstrate that native populations harbor nectar-free 'cheating' plants at low frequencies. BA emission however does not occur over the entire night, but is commonly released in a sharp peak in the first part of the night (*Kessler et al., 2010*), depending on the genotype and the environment. Hence, there are times in the night were one plant is emitting BA, while a neighboring plant is not in genetically diverse populations. Since nectar and scent production were uncorrelated in natural populations, we used RNAi to completely uncouple nectar and scent production in all combinations in an isogenetic background to rigorously examine their costs and benefits.

To investigate the consequences of a lack in nectar, floral scent, and both on pollination and herbivory, EV, SWEET9, CHAL, and CHALxSWEET9 plants were randomly planted 4 m apart in a field plot in the SW USA (Utah) in the plant's native habitat (*Figure 2A*). Since *N. attenuata* is a self-compatible plant, anthers were removed from five flowers per plant, and all additional flowers were removed. Hence, seeds of treated flowers were produced entirely from pollinator-mediated outcrossing with neighboring WT plants, which were planted between the rows of transformed plants and were allowed to flower normally. To evaluate the overall pollinator attractiveness of the lines, we allowed floral visitors free access to treatment plants for the flowers' lifetime (3 days). All three lines lacking either floral scent, nectar or both, produced significantly fewer seeds in comparison to similarly antherectomized EV plants (*Figure 2B*; *Friedman* signed rank test $\chi^2$ = 10.22, df = 3, n = 16–20, p = 0.017). CHAL plants produced only 22.9% (p < 0.001), Sweet9 9.7% (p < 0.001), and CHALxSWEET9 11.6% (p < 0.001) of the seeds produced by EV from outcrossed pollen. The most frequent floral visitors of *N. attenuata* at night in the field at the time of the experiment were *M. sexta*,

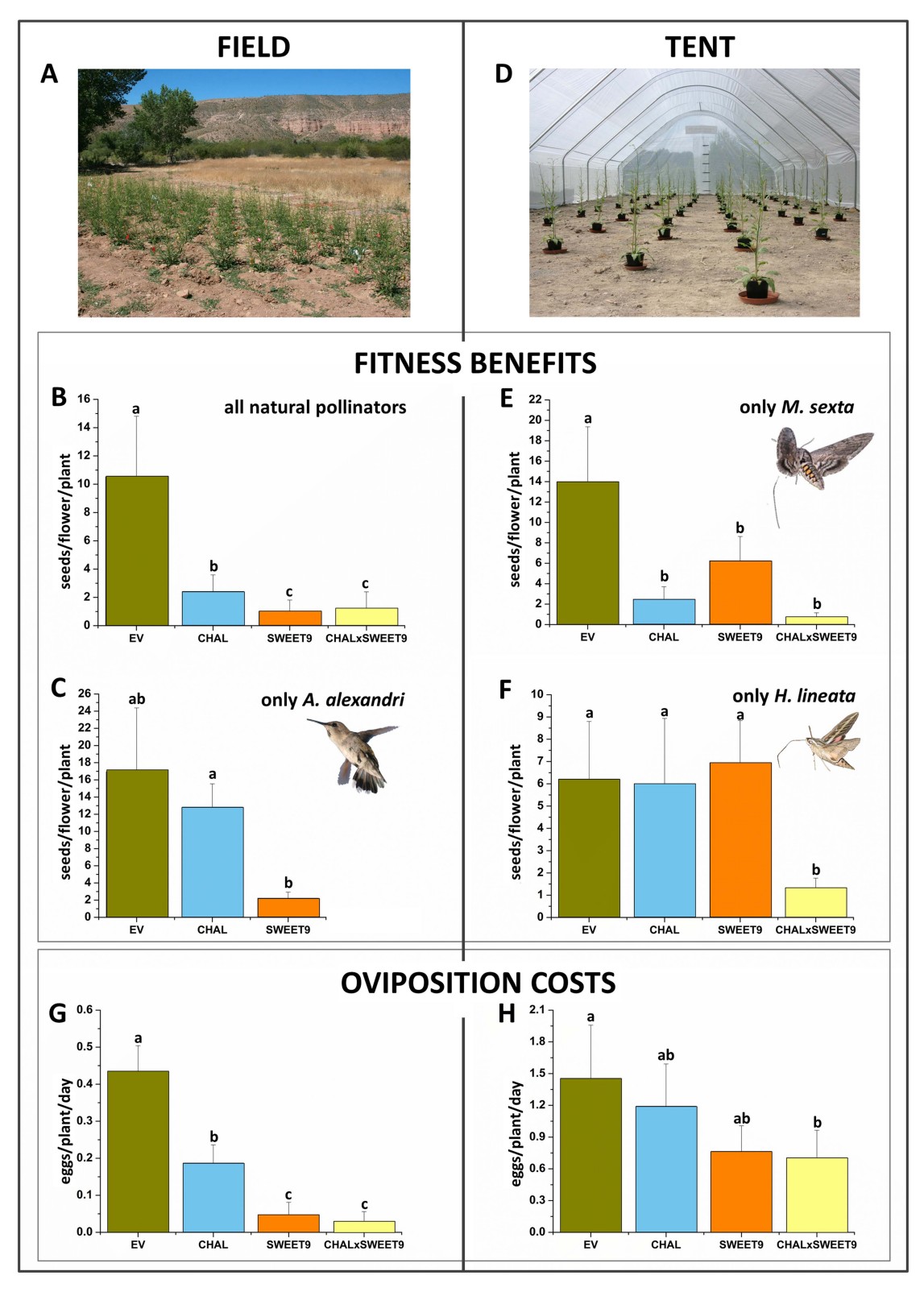

**Figure 2**. Benefits and cost of floral scent and nectar, as revealed from pollination and oviposition experiments in the field and tent. Means +SEM of either seed production of antherectomized flowers or oviposition by *Manduca sexta* on transformed plants silenced in the production of floral scent (CHAL), floral nectar (SWEET9), or both (CHALxSWEET9) in comparison to empty vector control plants (EV). (**A**) Field plot in *N. attenuata's* native habitat at the

*Figure 2. Continued*

Lytle ranch preserve in Santa Clara, Utah, USA. (**B**) Seeds sired from pollen transferred by the native community of floral visitors over the complete life span of a flower (3 days). (**C**) Exclusive pollination by *Archilochus alexandri* hummingbirds in the field during a 12 hr day. (**D**) Tent set-up used for single species pollinations in Isserstedt, Germany. Seeds sired from pollination of single *M. sexta* (**E**) or *Hyles lineata* (**F**) individuals during a 9-hr night. Eggs oviposited per transformed line (n = 10) on different days by the native community of *M. sexta* in the field (**G**) or by single individuals in the tent (**H**). Letters indicate significant differences inferred by a Friedman signed rank test p < 0.05.

The following figure supplements are available for figure 2:

**Figure supplement 1**. Nectar accumulation in flowers of transformed lines in the field.

**Figure supplement 2**. Analysis of flower volatiles.

**Figure supplement 3**. Principal components analysis (PCA) of leaf volatiles.

*Manduca quinquemaculata*, and *Hyles lineata* hawkmoths (Sphingidae), and *Archilochus alexandri* hummingbirds (Trochilidae) during the day time. In order to evaluate each species individually, we covered experimental plants at night with mesh cones (*Kessler et al., 2010*) to only allow hummingbirds to access the flowers. While CHAL (75.6%; *Friedman* signed rank test $\chi^2 = 7.66$, df = 2, n = 24, p = 0.02; p = 0.835) plants produced almost as many seeds as EV plant, SWEET9 plants produced considerably fewer seeds than did EV plants (13.1%; p = 0.074) and significantly fewer seeds than did CHAL plants (17.4%; p = 0.005; *Figure 2C*).

Of the three hawkmoth species occurring in the field, we examined two (*M. sexta* and *H. lineata*) in a tent set-up. To simulate field conditions for the moths, we used a mesh tent large enough for natural moth pollination behavior (24 m × 8 m; *Figure 2D*) and examined the consequences of pollination services from single hawkmoths of a given species at a time, something that was not possible to do in the field. To avoid inbreeding or artificial larval diet effects, only moths that had been reared from field-collected eggs and maintained on a plant diet were used.

We used a different moth on each experimental day. *M. sexta* strongly distinguished between EV and all other lines (*Figure 2E*; *Friedman* signed rank test $\chi^2 = 29.90$, df = 3, n = 30, p < 0.001). Although SWEET9 lines produced only 44.6% of the seeds produced by EV plants (p = 0.006), the avoidance of plants which produce no floral scent had an even stronger effect on seed set. CHAL plants produced only 17.4% of the seeds of EV (p < 0.001), just as CHALxSWEET9 plants which produced only 5.2% of the seeds produced by EV plants (p < 0.001). Interestingly, *H. lineata*, a second naturally occurring hawkmoth species, showed a different response to that of *M. sexta* (*Figure 2F*; *Friedman* signed rank test $\chi^2 = 12.35$, df = 3, n = 35, p = 0.006). Both CHAL (96.6%; p = 0.853) and SWEET9 (111.69%; p = 0.578) plants produced a number of seeds equivalent to those of EV plants. Only when the flowers lacked both floral traits (CHALxSWEET9) was seed set mediated by *H. lineata* pollinator activity significantly reduced (21.3%; p = 0.003).

Our results highlight the value of using both floral scent attraction as well as a reward in order to cope with an unpredictable community of pollinators. While floral scent is an essential floral cue for *M. sexta*, it is clearly not for *H. lineata*. While nectar is essential for pollination by *A. alexandri*, the lack of nectar does not reduce the pollination services provided by *H. lineata* visitations. Though more generalist pollinators such as *H. lineata* seem to pollinate even when scent or nectar is missing, they strongly reject plants which are unable to produce both floral traits. However, for the more specialized pollinator, *M. sexta*, both floral scent attraction and nectar reward are important to maximize outcrossing. Although all lines lacking a certain floral trait showed reduced seed set, plants unable to produce nectar still produced more than twice as many seeds than plants unable to produce floral scent. Again, we found a stronger negative effect in plants unable to produce both traits. In contrast, *A. alexandri*, the only daytime visitor, rejected plants which did not produce nectar, but surprisingly was also less attracted to plants unable to produce floral scent, although visiting flowers at times when BA is not emitted. One explanation for this could be the accumulation of floral scent in nectar (*Raguso, 2004*), which could influence the taste of the nectar for these pollinators (*Kessler and Baldwin, 2007*). Another hypothesis could be that hummingbirds also use olfaction and are able to

use nectar scent to locate a proper nectar source. Birds, long suspected to use scent in homing and navigation (*Keeton, 1974*) have recently been suggested to use scent for prey localization (*Mantyla et al., 2008*).

Hummingbirds, in contrast to hawkmoths, have smaller territories and pollinate flowers only in a small area close to their nests (*Norton et al., 1982*). This allows them to learn cues and plant positions which are correlated with the presence of food sources (*Ackerman et al., 1994*; *Campbell et al., 1997*; *Hurly and Healy, 2002*) or even with the quality of the food (*Pérez et al., 2011*). In contrast to hummingbirds, large hawkmoths like *M. sexta* cover much larger distances each night and rely more on direct perception rather than memory in visiting their food sources (*Raguso and Willis, 2003*, *2005*), as they most likely never visit the same plant twice in nature, although moths are also known to learn when exploring new food sources (*Riffell et al., 2013*). The inferences we can draw about learning in moths from our results are limited. The pollination assays were conducted with naive moths that lacked experience with rewardless and scentless *N. attenuata* plants. Experienced *H. lineata* might have behaved differently, as has been shown in orchid pollinating bumblebees which were able to learn to avoid deceptive orchids (*Internicola and Harder, 2012*). Our experiments did not directly measure the behavior of the floral visitors and data on the variance in seed set resulting from pollinator visitations, limits the inferences we can draw about pollinator behavior. While there was no evidence that pollinators can sense the presence of nectar, we cannot reject this hypothesis (*von Arx et al., 2012*).

For future research, it would be interesting to investigate how geitonogamy (*Fisogni et al., 2011*) would be influenced in rewardless plants. We measured only the pollen flow from one plant to another and not within a given inflorescence. The behaviors of the three tested pollinator species appear to be quite different, and these differences have the potential to influence pollen transfer within an inflorescence and thereby decrease the chance of receiving outcrossed pollen. *H. lineata*, *M. sexta*, and *M. quinquemaculata*, for example, show different patterns of movement in both the field and the tent. While *H. lineata* moves sequentially from one flower to the next—visiting all flowers within one inflorescence (even visiting the same flower multiple times)—and from one plant to the next plant, both *Manduca* species visit only a few flowers within an inflorescence before moving on and skipping as many as 10 intervening plants (personal observations).

*N. attenuata* is capable of producing the full set of seeds of comparable mass and number from self-pollination as from outcrossing (*Baldwin et al., 1997*). Hence, the plant can compensate for the lack of pollinators by selfing, and the function of producing floral nectar and scent is to ensure outcrossing to maintain genetic variability. Given that *N. attenuata* seeds can spend hundreds of years in the seed bank before germinating (*Preston and Baldwin, 1999*), outcrossing will likely be associated with longevity in the seedbank. In other systems outcrossing is thought to increase herbivore resistance in future generations (*Bello-Bedoy and Núñez-Farfán, 2011*).

How does the production of floral scent or nectar influence *M. sexta* herbivory? To answer this question, the number of flowers was reduced to the same number on all plants on a given day and oviposited eggs were counted the following morning. In the field, all lines lacking a certain floral trait had reduced loads of eggs (*Figure 2G*; *Friedman* signed rank test $\chi^2 = 11.15$, df = 3, n = 4, p = 0.0109). On CHAL plants, only 43.1% (p = 0.046) of the eggs found on EV plants were oviposited. The effect was even stronger in plants unable to produce nectar (SWEET9, 10.8%; CHALxSWEET9, 6.5%, p = 0.046). Comparable results were obtained in a similar experiment, conducted in the tent with single moths, (*Figure 2H*; $\chi^2 = 13.73$, df = 3, n = 11, p = 0.0033). CHAL plants received only 81.9% (p = 0.096) of the eggs of EV plants, while SWEET9 received 52.5% (p = 0.058) and CHALxSWEET9 received 48.4% (p = 0.007) of eggs compared with EV. To control for any other leaf-based influences on oviposition, we conducted control experiments with plants without flowers, by using non-flowering plants or plants from which flowers had been removed before the experiment. In the non-flowering stage, no differences in oviposition were found between the different lines (*Friedman* signed rank test $\chi^2 = 2.81$, df = 3, n = 6, p = 0.4214). In the field, oviposition rates (over eight experimental days) were too low for statistical analyses, when plants did not flower. Although, plants unable to produce floral scent had a significantly lower chance of receiving an egg from *M. sexta*, nectar seems to play a larger role in host choice. In contrast to the pollination results, we found no elevated effect in the CHALxSWEET9 cross compared to the single silenced line, which is consistent with the strong effect of nectar alone. The fact that plants lacking floral scent have reduced ovipositions as well as reduced seed set is probably due to the difficulties hawkmoths have in locating scentless *N. attenuata* flowers (*Kessler et al., 2008*).

It is intriguing that it is the reward more than the attractant which influences the female hawkmoth's oviposition decisions, while the attractant plays a similar role as the reward when the same species is acting as a pollinator. Of course our conclusions are context dependent, depending on the frequency of low-nectar or low-scented plants in a native population. Clearly, there needs to be an attractive component (e.g., floral scent) coupled with the nectar reward to allow moths to associate food sources with certain floral signals (*Daly and Smith, 2000*) or oviposition sites. Similarly, moths would learn to avoid floral advertisements if they are not associated with a reward (*Haber, 1984*; *Smithson, 2001*). In an evolutionary context, our findings suggest that for an interaction with *M. sexta*, *N. attenuata* plants should minimize their nectar accumulation while maximizing their floral scent emission in order to maximize fitness. Indeed *N. attenuata* produces tiny amounts of nectar in comparison to other sympatric plant species that compete for *M. sexta's* pollination services (*Raguso et al., 2003*; *Riffell et al., 2008*), and thus seems to have optimized the attractant–reward proportions to maximize its fitness when dealing with this pollinator which is also an herbivore, albeit at a different life stage. *D. wrightii*, for example, a plant which shares *M. sexta* as both pollinator and herbivore, is a perennial plant that accumulates 20 times more nectar, produces a large amount of foliage that allows it to better compensate for losses to herbivores, and recruits fewer pollinating species than does *N. attenuata* (*Kessler, 2012*).

Pollination success of *N. attenuata* is ensured by recruiting several pollinator species, which may influence the evolution of floral scent and nectar reward. In the absence of *M. sexta*, species such as hummingbirds which only pollinate plants capable of producing nectar, and probably will also only continue to pollinate if a sufficient quantity of nectar is available, are important pollen vectors. *H. lineata* also provides adequate pollination services, but without apparent evolutionary pressure on these *N. attenuata* floral traits, as this species seems to require neither an attractive signal nor a reward, at least if these plants are surrounded by other nectar and scent producing plants. Moreover *H. lineata* is also not an herbivore on *N. attenuata* during its larval stage. These two sphingid species show dramatic differences in their responses to both floral traits. Only *M. sexta* seems to rely on the cues provided by the flower and distinguishes between honest and cheating plants in a mixed *N. attenuata* population. For *H. lineata*, either floral scent or nectar is sufficient for the moth to provide full pollination services. In bee pollinated *Brassica rapa* plants, the honesty of a floral signal—that a certain scent would consistently lead to a reward—plays a key role in the plants attractiveness to pollinators (*Knauer and Schiestl, 2015*), something which seems not to be the case for *N. attenuata* and its pollinators, as we found no association between BA emission and nectar volume, or nectar sugar concentration. Our data are consistent with a trade-off between rewarding pollinators and avoiding herbivores in the evolution of floral nectar in *N. attenuata*, but why is BA emission so variable? Since antagonists also use floral scent to locate host plants, increasing floral scent emissions would not only increased *M. sexta* pollination but could also incur costs by possibly increasing florivory or nectar robbing. Carpenter bees, for example, tend to rob more *N. attenuata* flowers on plants that emit BA than from plants that do not (*Kessler et al., 2008*) and experimentally enhancing BA emissions increased rates of browsing (*Baldwin et al., 1997*).

In this study, all pollinators visited *N. attenuata* at different times between dusk and dawn and it is an interesting question how this temporal order of nectar removal influences the pollination services they provide for the plants. *A. alexandri* visits *N. attenuata* approximately 2 h before dusk until dusk and again in the morning after dawn for 2 h. *H. lineata* visits from 1 h before dusk until 1 h after dawn, and finally both *Manduca* species pollinate and oviposit after dusk until dawn, with periods of enhanced activity depending on moonlight, temperature, and other conditions. We had expected that these different visiting times would blur the differences in the seed set amongst the lines differing in scent and nectar production when plants were exposed to the entire community of floral visitors at the field station, but the results were just the opposite. The differences between EV plants and SWEET9 as well as CHAL plants were maximized when plants were exposed to the entire community of pollinators at once, suggesting that the behavioral responses of each different pollinator to a given floral trait reinforce each other in ways that can only be understood through more detailed behavioral observations. *N. attenuata* flowers secrete nectar slowly over the night until approximately 4 am (*Kessler, 2012*), theoretically keeping a low but constant amount of nectar over the entire night, even if floral visitors removed the majority of nectar in the dusk or in the first half of the night. Unfortunately, we do not know if nectar secretion patterns change after flowers have been visited and nectar has been removed, but from nectar measurements, we know that flowers visited by *M. sexta*

(which invariably removes the standing volume of nectar) again contain a residual amount of nectar in the morning. Similar questions remain about the costs associated with the production of nectar. One would expect that nectaring hummingbirds or white-lined sphinx moths in the dusk could influence *M. sexta* oviposition by removing all the nectar in the 2 h before *M. sexta* becomes active. Clearly, there remains a large gap between our understanding of how particular flower traits influence the behavior of the flower visitors and the pollination services that they provide.

Our model system appears particular in that pollinators are at the same time herbivores, but this ecological dilemma is not uncommon for many plant species (*Irwin, 2010*), and the lessons we draw from our study are likely applicable for other plant systems with differently structured interactions with insects. For example, in the *Silene latifolia–Hadena bicruris* nursery pollination system, male moths provide a fitness benefit for the plant, while interactions with females come with the substantial costs of seed-feeding caterpillars (*Labouche and Bernasconi, 2009*). The highly specialized *Yucca-Tegiticula* pollination/seed predation system would be another example. Plants rarely interact with just one partner and usually face multiple selection pressures from mutualists and antagonists at the same time, which may shape the different floral traits of a plant. In most cases, there will be less and more effective pollinators (*Conner et al., 1995*; *Barthelmess et al., 2006*; *Matsuki et al., 2008*; *Miller et al., 2014*), as well as herbivores which are attracted by floral signals such as scent or nectar (*Adler and Bronstein, 2004*; *Theis, 2006*). Even purely mutualistic interactions like those known from bee pollination systems, probably evolved in the context of avoiding herbivory and excluding less efficient pollinators at the same time. Pollinator networks are known to be highly flexible and change frequently over time within a season (*Olesen et al., 2008*), as well as between years (*Olesen et al., 2011*) and this flexibility likely explains why plants recruit additional pollinator species rather than specialize if they rely on outcrossing. Most pollinator networks are nested (*Pawar, 2014*) and this nestedness is thought to stabilize mutualistic networks (*Rohr et al., 2014*), reducing interspecific competition and enhancing the number of coexisting species (*Bastolla et al., 2009*), with the end result of increasing the coexistence of several pollinating species on a plant. The high temporal plasticity in species composition coupled with the low variation in network structure properties in pollination networks makes such a scenario likely (*Petanidou et al., 2008*).

Flowers face a multidimensional challenge. They have to ensure outcrossing by an unpredictable number of pollinating species and individuals, all of which have different preferences and behaviors, and at the same time need to keep herbivores at bay. Here, we demonstrate that nectar-free, pollinator cheating, varieties of *N. attenuata* occur in nature, and that producing no nectar can substantially reduce ovipositions by *M. sexta*, but that this comes with a reduction in pollination services. While cheating on the nectar rewards works for one pollinator species, it does not for others. In contrast, cheating with regards to the attractant floral scent has little influence on *M. sexta* oviposition, but dramatically reduces outcrossing by *M. sexta*. All three examined pollinators show different responses to nectar and scent and thus the interactions of multiple species are likely responsible for fine-tuning the evolution of both nectar and floral scent, as both *H. lineata*, as well as *A. alexandri* are important pollen vectors in the absence of *M. sexta*. Both traits in combination are required to maximize maternal fitness in nature particularly in ensuring *M. sexta* visitations, which we hypothesize delivers the greatest diversity of pollen genotypes compared to *H. lineata* and *A. alexandri*. Hence, the close association between *N. attenuata* and *M. sexta* may result from *M. sexta's* ability to move pollen over longer distances than the other pollinators, and thus provide superior outcrossing services among *N. attenuata* populations which are frequently isolated by the large distances that frequently occur between fires, and hence *N. attenuata* populations. The long time between fires at any particular location may place a premium on outcrossing rates, as heterozygosity may increase herbivore resistance in future generations (*Mescher et al., 2009*; *Bello-Bedoy and Nunez-Farfan, 2011*), as well as longevity in the seedbank. We conclude that herbivores, as well as pollinators shape the evolution of floral traits and that it makes little sense to study floral traits as if they only mediate pollination services.

## Materials and methods

### Plants, growth conditions, and field plantations

Wild-type (WT) *N. attenuata* plants, obtained from seeds collected from a native population in 1988 at the DI Ranch (Santa Clara, UT) and subsequently inbred for 22 or 30 generations, were transformed

with *Agrobacterium tumefaciens* (strain LBA 4404) (*Krügel et al., 2002*) containing the construct pRESC5CHAL to silence *N. attenuata* chalcone synthase (*Kessler et al., 2008*), or the construct pSOL8SWEET9 to silence *N. attenuata* sweet9 (*Lin et al., 2014*). The vector construction and transformation procedures have been previously described (*Krügel et al., 2002*). For both CHAL and SWEET9, we choose one line which harbored only one single insertion of the construct and which had completely normal growth and the strongest reductions in either floral volatile or nectar production. Both lines have been fully described previously: CHAL A-07-283-5 (*Kessler et al., 2008*), SWEET9 A-10-198-4 (*Lin et al., 2014*). To create a hemizygous cross between CHAL and SWEET9, anthers of CHAL flowers of the T2 generation were removed and flowers hand pollinated with pollen of a SWEET9 (T2 generation) plant. The resulting T3 crossed seeds were used in all field and tent experiments together with the T3 generations of CHAL and SWEET9 plants. As a control, we used EV line A-04-266-3 transformed with pSOL3NC (*Bubner et al., 2006*), which is known to be completely comparable to wild-type plants (*Schwachtje et al., 2008*).

Seed germination was performed in both glasshouse and field as described by *Krügel et al. (2002)*. For tent or glasshouse experiments, plants were grown individually in 1 (glasshouse) or 2 L pots (tent) in the glasshouse at 26–28°C under 16 h of light supplied by Philips SON-T Agro 400 (Philips, Germany, http://www.lighting.philips.co.uk/) sodium lights until used for experiments in glasshouse or tent. The tent (Amiran, Kenya, www.amirankenya.com) has a dimension of 8 × 24 m and is about 4 m high. With a distance of 2 m among plants, 60 plants can fit into the tent. The sides and the front the tent is covered with mesh, which allows for air exchange. In the tent, plants were positioned 2 m apart from each other in rows, with three different transformed plants in one row, followed by a row of three WT plants, which served as pollen donors. In total, five plants per transgenic line were positioned in the tent per night for pollination experiments or 10 plants per transgenic line were used for oviposition trials.

In the field, seedlings were transferred into previously hydrated 50-mm peat pellets (Jiffy 703, Always Grows, Sandusky, OH, http://www.alwaysgrows.com/) 14 days after germination and were gradually adapted to the environmental conditions of high sun and low relative humidity of the Great Basin Desert habitat over 14 days by keeping the seedlings in the shade. Adapted size-matched seedlings were transplanted into the field plot at the Lytle Ranch Preserve (Santa Clara). Seedlings were watered every other day until roots were established. The four different transformed genotypes were randomly planted in rows with 4 m between each other and with one pair of WT plants between two transformed plants, which served as pollen donors in pollination experiments. Seeds of the transformed *N. attenuata* lines were imported under US Department of Agriculture Animal and Plant Health Inspection Service (APHIS) notification numbers 07-341-101, 10-349-101m, and 11-350-102m and the field experiments were performed under notification numbers 06-242-03r, 10-349-102r, and 13-350-101r.

To characterize the phenotypes of all transformed lines, nectar accumulation (*Figure 2—figure supplement 1*), nectar sugar concentration, floral scent emission (*Figure 2—figure supplement 2*), as well as leaf volatile emissions (*Figure 2—figure supplement 3*) were measured in the field. While there were no differences in leaf volatile emission and nectar sugar concentration among the four lines, the lack of nectar in Nasweet9-silenced lines (SWEET9 and CHALxSWEET9), as well as the lack of floral BA emission in Nachal1-silenced lines (CHAL and CHALxSWEET9) were confirmed. No phenological differences, such as plant shape, size, or flower size among these four lines, would suggest a lack of fitness benefit for plants producing no nectar or no scent. In the case of the hemizygous cross between CHAL and SWEET9 (CHALxSWEET9), we observed that occasionally plants started to secrete nectar in late developmental stages. This phenomenon is likely caused by methylation events which occur in aging hemizygous lines (*Weinhold et al., 2013*). In all experiments, we therefore measured nectar production daily and excluded plants which had lost their initial no-nectar phenotype.

## Nectar collection

All flowers were removed daily before opening. On experimental days, flowers were allowed to open in the evening and inflorescences were covered with plastic bags (Plastibrand, Wertheim, Germany) to exclude pollinators and to retard nectar evaporation. Nectar volume was measured with 25 µL glass capillaries (BLAUBRAND, Wertheim, Germany) on the following morning between 5 and 6 am, at the

time of the maximal nectar accumulation (*Kessler, 2012*). To quantify nectar accumulations, the entire corolla was removed with mild pressure from the flower. The entire nectar remains in the corolla tube and can be squeezed from the tube into a glass capillary (*Rothe et al., 2013*). Nectar in the capillary was quantified with a ruler. To quantify nectar sugar concentration, we used a portable refractometer (Optech, Sliedrecht, the Netherlands) with a range from 0 to 32% and a resolution of 0.2%. Nectar of 4–7 flowers per plant was measured to assess the variability in the different native phenotypes (single plants), and two flowers per plant were measured in 10 plants of each of our transformed lines, as well as the nectar-free natural phenotypes. For statistics, the average nectar volume or nectar sugar concentration of all measured flowers of a plant was used.

## Analysis of plant volatiles from leaf and flower headspaces

Plant volatiles were collected on silicone tubing (ST) and analyzed as described in *Kallenbach et al. (2014)*. Briefly, polyethylene terephthalate (PET) containers were used to enclose leaves (300 mL sampling volume) and flowers (30 mL sampling volume) for headspace sampling. Leaf volatiles were collected in the glasshouse overnight using a similar, fully expanded, mature, non-senescent stem leaf from each plant (n = 7, per genotype). Flower volatiles from EV, CHAL, SWEET9, and CHALxSWEET9 (*Figure 2—figure supplement 2*) were collected on ST as described (*Kallenbach et al., 2014*) in the field by trapping flowers overnight (8 pm–6 am) on the first night of opening. Flower volatiles from native populations (*Figure 1*) were collected using Super-Q traps and analyzed as described in *Wu et al. (2008)*. Empty trapping containers distributed among plants were used as background controls. BA levels were calculated using external calibrations.

Volatile analysis was performed on a TD-20 thermal desorption unit (Shimadzu, Germany, www. shimadzu.de) connected to a quadrupole GC-MS-QP2010Ultra (Shimadzu). Individual STs were placed in 89 mm glass thermo desorption (TD) tubes (Supelco, Germany, sigmaaldrich.com) and desorbed under a stream of nitrogen at 60 ml min$^{-1}$ for 8 min at 200°C. Desorbed volatiles were cryo-focused at −20°C onto a Tenax adsorbent trap in front of the column. After desorption, the Tenax trap was heated to 230°C within 10 s, and analytes were injected with a 1:20 split ratio onto a ZB-Wax-plus column (30 m long, 0.25 mm i.d., 0.25 μm film thickness; Phenomenex, Germany, www.phenomenex.com) with He as the carrier gas at a constant linear velocity of 40 cm s$^{-1}$. The TD-GC interface was maintained at 230°C. Two different Gas Chromatograph (GC) oven gradients for the profiling of leaf and flower headspaces were used. For leaf samples, the oven was held at 40°C for 5 min, then ramped to 185°C at 5.0°C min$^{-1}$, and finally to 230°C at 30°C min$^{-1}$, where it was held for 0.5 min. For analysis of flower samples, the oven was held at 60°C for 1 min, then ramped to 150°C at 30°C min$^{-1}$, then to 200°C at 10°C min$^{-1}$, and finally to 230°C at 30°C min$^{-1}$ and held for 1 min. Electron impact (EI) spectra were recorded at 70 eV in scan mode from 33 to 400 m/z using a scan speed of 2000 Da s$^{-1}$. The transfer line was maintained at 240°C and the ion source at 220°C.

Data processing and export were performed using the Shimadzu GCMS solutions software (v2.72). Flower volatiles were analyzed against our reference standards library. For leaf volatiles analysis, raw data files were converted to netCDF format and processed using the R (http://www.r-project.org) packages XCMS (*Tautenhahn et al., 2008*) and CAMERA (http://www.bioconductor.org/biocLite.R) as described (*Gaquerel et al., 2010*). After removal of contaminants and known artifact peaks, 94 compounds were detected, and the Metaboanalyst software (5, 6) was used to perform principal components analysis (PCA).

## Pollination studies

As *N. attenuata* is a self-compatible plant, anthers of flowers were removed in the morning between 5 and 7 am from experimental flowers, just before anthesis (*Kessler et al., 2008*) and 12 hr before flowers open and are accessible for pollinators. Five flowers per plant were antherectomized. All other floral buds which were about to open within the next 2 days were removed. WT plants located between the experimental plants of the different genotypes were allowed to flower and served as pollen donors. This procedure was used in both tent and field.

In a control experiment in the tent, all four lines used in the experiments were equally able to produce seeds from hand-pollinated flowers (one-way ANOVA: $F_{3,\ 36} = 1.19$, p = 0.33; n = 10). EV plants produced on average 233 ± 23 seeds per flower, CHAL 242 ± 22, SWEET9 209 ± 21, and CHALxSWEET9 265 ± 19. We have no data available for the longevity of seeds of the four lines in the seedbank, which would allow predictions about fitness benefits for future generations of these lines. As such data require years to collect, it is beyond the scope of this study.

In a 2011 field experiment, the response of hummingbirds was tested without including the CHALxSWEET9 cross, as these seeds were not available at this time. To exclude night-time pollinators and thus to evaluate only hummingbird pollination, plants were covered with mesh cones (*Kessler et al., 2010*) between 8 pm and 6 am. By visual observation, it was ensured that only *A. alexandri* hummingbirds were visiting the flowers during the times when flowers were accessible to visitors. Bees were not present on the field plot at the time of hummingbird experiments on 2 days in 2011. Experiments planned to investigate the pollination services of the entire community of floral visitors in 2012 and 2013 failed due to fungal or bacterial diseases which killed plants before they started to flower. In 2014, we conducted an experiment, in which we allowed the entire community (day- and night-pollinators) of floral visitors to access treated plants on four experimental days. All plants in this experiment were accessible for floral visitors over the entire lifespan of the atherectomized flowers (3 days). All these experiments have not been influenced by nectar robbing from carpenter bees (*Xylocopa* spp.). On all experimental days, we monitored their presence, and experiments were not conducted if carpenter bees occurred on the plot, which happened in the later parts of the 2011 and 2014 seasons, when pollination experiments had to be stopped.

In the tent, plants were treated similarly to those of the field experiments, and one single male *M. sexta* or *H. lineata* was released in the tent to interact with the experimental plants overnight. Only moths raised on *N. attenuata* or *N. tabacum* (*M. sexta*) or *Fuchsia* sp. (*H. lineata*) from field-collected eggs were used for experiments in the tent. In the following morning, plants were moved from the tent into the glasshouse until capsules had matured (approximately 14 days). On each experimental day, new plants and a new moth were used. In total seven *M. sexta* and seven *H. lineata* moths were tested in these experiments.

## Oviposition trials

The same plants used for pollination experiments were used for *M. sexta* oviposition in the field. Flower number of all plants was reduced to five newly opening flowers every afternoon so as to standardize the apparency of each plant's inflorescence to the moths. Each morning, eggs were counted and removed from the plants. The collected eggs, as well as eggs collected from *D. wrightii* plants at this time of the season were used to rear moths for experiments in the tent. All caterpillars which hatched from these eggs proved to be *M. sexta*, which suggests that *M. sexta* accounts for all the oviposition data in our experiments, and not *M. quinquemaculata*, a closely related species which shares host plants with *M. sexta* in both the larval and the adult stages. Of the nine experimental days, *M. sexta* moths oviposited on only four of these days. In the tent experiments, unlike in the pollination trials, no WT pollen donor plants were necessary, which allowed for a larger number of experimental plants in the tent at a time. 10 plants per transformed line were randomly arranged in the tent with a distance of 2 m between plants. Plants were exchanged and rearranged daily, and a freshly mated female *M. sexta* moth was released into the tent on each day.

## Statistic procedures

For all pollination assays, experiments were conducted over several days. In all cases, the day did not influence the line effect significantly as revealed by a one-way ANOVA p > 0.05. Therefore, each plant was used as a replicate and the average seed number produced from the five flowers on a plant was used for statistical analyses. For oviposition data, days were used as replicates and average values of the different lines were compared. Significant differences for all pollination, oviposition, as well as phenotyping data (nectar volume and floral volatile emissions) data were calculated using a *Friedman* signed rank test which was performed in R2.11.1 (*Crawley, 2007*; *R Development Core Team, 2007*). Control hand-pollinations were compared using a Fisher's PLSD *post-hoc* test following a one-way ANOVA, which were performed using StatView program version 5.0 (SAS Institute Inc., NC, USA; www.statview.com).

## Acknowledgements

This work was funded by the European Research Council advanced grant ClockworkGreen (no. 293926) to ITB, the Global Research Lab program (2012055546) from the National Research Foundation of Korea, and by the Max-Planck-Society. We thank Brigham Young University for the use of the Lytle Ranch Preserve field station, USDA-APHIS for constructive regulatory oversight of the GMO releases, Theresa Erler, Meredith C Schuman, and Antje Wissgott for help with the tent experiments, and the Utah field crew 2014 for help with the emasculation of flowers.

## Additional information

### Competing interests

ITB: Senior editor, *eLife*. The other authors declare that no competing interests exist.

### Funding

| Funder | Grant reference | Author |
|---|---|---|
| Max-Planck-Gesellschaft (Max Planck Society) | | Ian T Baldwin, Danny Kessler, Mario Kallenbach, Eva Rothe, Celia Diezel, Mark Murdoch |
| European Research Council (ERC) | Advanced Grant no. 293926 | Ian T Baldwin, Celia Diezel, Mario Kallenbach, Eva Rothe |
| National Research Foundation of Korea (NRF) | Global Research Lab program (2012055546) | Ian T Baldwin |

The funders had no role in study design, data collection and interpretation, or the decision to submit the work for publication.

### Author contributions

DK, MK, ITB, Conception and design, Acquisition of data, Analysis and interpretation of data, Drafting or revising the article; CD, ER, MM, Acquisition of data, Analysis and interpretation of data, Drafting or revising the article

### Author ORCIDs

Ian T Baldwin, http://orcid.org/0000-0001-5371-2974

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
