## [Decision Letter]

Thank you for sending your work entitled “How scent and nectar influence floral antagonists and mutualists” for consideration at *eLife*. Your article has been favorably evaluated by Detlef Weigel (Senior editor), a Reviewing editor, and three reviewers, one of whom, Ian Kaplan, has agreed to share his identity.

All three reviewers praise the experiments and find your results (some of which are quite surprising) very interesting. The work provides a very nice contribution to this field of research. You will find the specific comments of the reviewers below.

The main issues to be addressed are the following:

1) Connection to the literature: the reviewers were unanimous in commenting that they feel that previous literature can be referred to in a more respectful way. Moreover, the reviewers would like to see more focus in the Discussion on the wider applicability of the results: are they an oddity of this system, or are they likely to occur in other systems as well; the *Yucca* system is mentioned, but one can also think of the *Silene-Hadena* system. Even when data are not as extensively available, the natural history aspects of the systems may allow predictions.

2) The data on BA emission should be presented in absolute amounts such as ng/flower instead of peak areas that do not allow comparison with other studies. Providing emission rates in absolute amounts is standardly done.

3) Please discuss that the final outcome of fitness remains to be investigated, in the context of whether the herbivores feed on fruits and seeds.

4) Make sure that all statements are supported by statistical analyses.

Reviewer #1:

This study uses RNAi to silence specific scent and nectar production in *N. attenuata*, and studies the fitness effects in term of pollination and oviposition by a herbivore. The approach chosen is clearly excellent, and the results are very interesting. The study shows that it is not necessarily possible to predict the impact of the total pollinator community from studying individual pollinators. From the results with *H. lineata*, *M sexta* and the hummingbird one should expect it would be ideal for the plants to become scentless, as only *M. sexta* was strongly affected by scentless flowers, and this pollinator is also the specialised herbivore, which the plant may want to avoid anyway. For the whole community, however, reduced scent resulted in strongly reduced seed set. A weakness of the study is that final fitness outcomes of herbivory were not estimated, as herbivory on fruits and seeds was not investigated. A recent study in Brassica has shown that specialised herbivores do not necessarily feed on seeds (Lucas-Barbosa et al. 2013). Also, it should be outlined that the results found are quite specific for systems were pollinators are at the same time herbivores, and may not be broadly applicable to other pollination system such as bee pollination. This point should be made clear at some point in the paper. There are also a lot of errs and other problems in the manuscript as outlined below. The Result and Discussion section is broadly mixed with text belonging to the Materials and Methods section. The Introduction has a slightly arrogant tone at some places, sounding as if most previous studies investigating similar topics were inferior to this study, this is inappropriate given the fact that each study, including the one by the authors, has its strengths and weaknesses.

Introduction: “as with floral scent, the function of nectar on plant fitness has been studied in only a few plant systems”. Fitness effects of nectar has indeed been studied in many systems by adding nectar to rewardless flowers.

Introduction: “again this single floral trait was studied only in the context of a single insect-plant interaction under glasshouse conditions”. Again, many of the studies that added nectar to flowers investigated the trait in the natural pollinator community context.

Introduction: “attractive function of flowers”. This term is not appropriate here and at other places, because nectar is also a part of it. Use “floral signals” instead.

Results and Discussion: “these results demonstrate that native populations harbor nectar-free ‘cheating’ plants at low frequencies”. But the mechanism for the maintenance is not shown; the nectar-free individuals may just be spontaneous mutants with very low fitness. For a frequency dependency scenario, as outlined in the introduction, measurements of fitness in the nectarless individuals are necessary.

Results and Discussion: “large hawkmoths like *M. sexta* cover much larger distances each night and rely on direct perception rather an memory in visiting their food sources”. Moths also use learning when exploring new food sources, see e.g. (65).

Results and Discussion: “it is intriguing that it is the reward rather than the attractant that influences the female hawkmoth’s oviposition decisions”. Reward is also an attractant, actually it is the primary attractant for pollinators.

Figure 1 is very confusing: there are three different axis labels, but only two different dots – where is nectar sugar concentration indicated? Normally, different axis labels in a scatter plot show the association of two traits in the same individuals – here it is seemingly two traits that are shown independently, thus it would be better to use two separate graphs. For the inset it says: “Mean (+/-SE) standing nectar volume in newly opened flowers at the end of the nectar production period (5-6am) in three native phenotypes (Ut-WT, 83, and 84) found within a separate sample of 424 native *N. attenuata* plants that were screened for the presence of nectar.” Now does the graph show nectar amounts of three individuals? Why are there error bars for one individual? Does the statistical test compare individuals? For nectar measurements, standardise to amounts per individual flower.

For BA emission, give absolute amounts in ng/flower, not peak area. Using peak area you may reach wrong conclusion regarding the differences in emission (Results and Discussion, first paragraph), as dose response curves in a mass selective detector for any compound may not be linear. It also makes the scent amounts non-comparable to other studies, because different MSD detectors may have different dose/response ratios.

References:

Lucas-Barbosa D., van Loon J.J.A., Gols R., van Beek T.A. & Dicke M. (2013) Reproductive escape: annual plant responds to butterfly eggs by accelerating seed production. Functional Ecology 27, 245-54.

Riffell J.A., Lei H., Abrell L. & Hildebrand J.G. (2013) Neural Basis of a Pollinator's Buffet: Olfactory Specialization and Learning in *Manduca sexta*. Science 339, 200-4.

Reviewer #2:

I really enjoyed reading this excellent study. The results were quite interesting and have important implications for those working on plant-insect interactions. All of my comments are on presentation and say nothing about the methodological accomplishments of this work and the data presented.

The Introduction could use some reworking. The first paragraph has a weak transition into floral scent and introduces florivores too early (“Floral scent plays a central role in attracting both insect pollinators and florivores to flowers”), and awkwardly brings in species differences (“Single traits (e.g. single floral volatile compounds) are thought to influence individual pollinating species differently”). In these first two paragraphs you discuss scent and nectar independently, but avoid discussing studies that have manipulated the two traits together and theory underlying why the two should synergize (or not), which is far more relevant to your work. The paragraph starting “The influence of nectar quantities on herbivory…” – the focus/purpose of this section is unclear. It introduces cheating, but in an awkward way and seems to cover a lot of unrelated ground. This should/could be tightened up. I'm unclear why rare nectar-free plants would trick pollinators, but not herbivores as well. Some of these ideas could be fleshed out much better in this paragraph.

Your discussion of biological effects relative to your statistics seems overly loose. In several cases you make strong statements about effects that are not supported by your stats. For example: “for the more specialized pollinator, *M. sexta*, floral scent attraction is more important than the nectar reward”. But on Figure 2, both of those treatment bars have a ‘b’ over them meaning that the two treatments are the same. Another example: “It is intriguing that it is the reward rather than the attractant that influences the female hawkmoth's oviposition decisions”. This is not true – on Figure 2, both reward and attractant had strong effects on oviposition. In most of these cases you're referring to magnitude differences, some of which are not statistically supported, so you need to tighten up your language throughout when discussing biological effects relative to what your figures/stats show.

Another related statistical term issue, you misuse the term ‘additive’ in a few spots (“Again, we found an additive negative effect in plants unable to produce both traits” and “In contrast to the pollination results, we found no additive effect in the CHALxSWEET9 cross”). Most people would refer to additivity in terms of whether the effect of treatment A+B is the sum of the individual effects from A and B tested separately. I don't think this is how you're using it. Unfortunately you can't really test for additivity very well in many of these figures because your individual manipulations of scent and nectar have such strong effects on seeds/eggs that the expected additive outcome would a negative number!

One last statistical comment – it might be easier to analyze your data as a two-way ANOVA rather than a one-way because then you can statistically disentangle the main effects of scent and nectar and their interaction effect, since this is a fully crossed design.

You might add more consideration in the Discussion to your manipulations relative to nature. Although you establish the existence of nectar-less flowers, there were no scent free flowers in the wild so your manipulation for that trait is a theoretical assessment, which is different, or biologically could inform the consequences of plants lowering floral emission.

Can you comment on the physiological costs of your manipulation to plants? You focus your study on the ecological costs of flowers in terms of herbivore attraction, but people will also be interested in the direct costs, i.e., Do your manipulated lines show any evidence of enhanced growth, flower production, etc. when volatiles/nectar are silenced? These traits are presumed to be quite costly so their silencing must benefits plants in some way.

This is not a criticism of how you performed the study – I like the anther removal – but I'd be very curious to see your study repeated without the removal. Do you have any idea how your data would look on normal plants? In other words, can tobacco selfing entirely negate the fitness costs of reduced pollinator visits you observed in Figure 2?

End of Results and Discussion: “We conclude that both pollinators, as well as herbivores shape the evolution of floral traits and that it makes little sense to study floral traits as if they only mediate pollination services”. I don't disagree with your conclusion here, I'm just wondering if you can make a broader case for the importance of floral traits for herbivores here or elsewhere. Part of me feels like the *Manduca* system is unique because the pollinator and herbivore are one and the same like in *Yucca*. I know this is not necessarily true, but I feel like you can make a stronger case for the relevance of this to many (most?) plant-insect systems for those who might believe it's an oddity to the *Manduca-*tobacco interaction.

Reviewer #3:

This study combines several tools and ideas to address a problem that has been gathering steam in the study of floral ecology and evolution: how do selective pressures shape floral display and reward when all visitors (good, bad and indifferent) are accounted for? And how can manipulative experiments be used to ask such a question when both tools (and hypotheses) have been crude? The authors suggest that the tradeoff between sufficient pollinator services and herbivore avoidance might be managed by variation in floral scent and nectar, and then survey geographic / population level variation in these traits by growing seeds from different lines in a common garden, demonstrating log normal variation in benzyl acetone (the primary chemical pollinator attractant) and nectar (the primary reward) in nature. Then, they combine RNAi lines that silence benzyl acetone production, nectar production, or both and measure the reproductive consequences in arrays of antherectomized plants (which cannot self-pollinate) in field and flight cage experiments.

The results are quite exciting and in some cases, unexpected. Both scent and nectar are required to match the full levels of seed fitness observed in nature, when plants are exposed to the full visitor community. Hummingbirds alone required nectar to provide sufficient pollination services, although (mirroring previous results by these authors) removing benzyl acetone from nectar did reduce their effectiveness. Similarly, *Manduca sexta* hawkmoths required nectar to elicit full levels of oviposition on *N. attenuata* plants. However, as pollinators. *M. sexta* required both scent and nectar, whereas *Hyles lineata* moths required scent or nectar to render sufficient pollination services.

These experiments are not only elegant, but they are very difficult to accomplish as a choreography (germinate the plants, fly across the planet, make sure the RNAi worked, make sure they don't die, get the timing right for the natural visitors). Buried in the Methods are references to setbacks and lost field seasons, which underscore the degree of difficulty behind these assays. I have several questions about methods and presentation which can easily be dealt with by the authors. My other points are primarily requests that they modify some of their statements about prior studies on floral cheating. Again, I think these are easily dealt with, but I would like to see less dismissiveness in the authors' tone – empty flowers are a very old idea in pollination biology, and there is a large body of theoretical and experimental work on floral enemies shaping floral ecology and evolution. The beauty of this study, as in previous studies by these authors, is in its surgical manipulation of the traits in question and the inclusion of several floral visitors, some detrimental, in the design, with real fitness consequences as the response variables.

---

## [Author Response]

*1) Connection to the literature: the reviewers were unanimous in commenting that they feel that previous literature can be referred to in a more respectful way. Moreover, the reviewers would like to see more focus in the Discussion on the wider applicability of the results: are they an oddity of this system, or are they likely to occur in other systems as well; the* Yucca *system is mentioned, but one can also think of the* Silene-Hadena *system. Even when data are not as extensively available, the natural history aspects of the systems may allow predictions*.

Thank you for pointing this out. We recrafted most of the Introduction to more thoroughly acknowledge previous publications. We have incorporated the ideas suggested by the reviewers and added the literature they suggested. We have also added a paragraph to the Discussion which addresses the wider applicability of our results.

*2) The data on BA emission should be presented in absolute amounts such as ng/flower instead of peak areas that do not allow comparison with other studies. Providing emission rates in absolute amounts is standardly done*.

The data on BA emission are now presented in absolute amounts – ng/flower, as calculated from an external calibration curve.

*3) Please discuss that the final outcome of fitness remains to be investigated, in the context of whether the herbivores feed on fruits and seeds*.

In nature when a *N. attenuata* plant is infested with a *M. sexta* larva that remains on the plant for the duration of its development, the plant usually loses its entire fitness, as a single *M. sexta* caterpillar can consume the entire plant (stalk leaves, flowers, capsules…), leaving only a few rosette leaves. Only in the unlikely event of a summer rain, plants are usually unable to compensate for these losses in photosynthetic tissue by producing a second round of flowers, and thus seeds. Mimicking *M. sexta* damage by removing all stalk leaves leads to a strong reduction in flower number, with size-matched undamaged plants producing about five times more flowers. The main seed feeder in this system is the negro bug (*Coremelina* spp), and previous work (4) demonstrated the floral scent, BA, is not used by this bug as a host location cue. We added this point to the Results section.

*4) Make sure that all statements are supported by statistical analyses*.

Statements which have not been supported by statistics have been removed throughout the manuscript.

Reviewer #1:

*This study uses RNAi to silence specific scent and nectar production in* N. attenuata*, and studies the fitness effects in term of pollination and oviposition by a herbivore. The approach chosen is clearly excellent, and the results are very interesting. The study shows that it is not necessarily possible to predict the impact of the total pollinator community from studying individual pollinators. From the results with* H. lineata*,* M sexta *and the hummingbird one should expect it would be ideal for the plants to become scentless, as only* M. sexta *was strongly affected by scentless flowers, and this pollinator is also the specialised herbivore, which the plant may want to avoid anyway. For the whole community, however, reduced scent resulted in strongly reduced seed set. A weakness of the study is that final fitness outcomes of herbivory were not estimated, as herbivory on fruits and seeds was not investigated. A recent study in Brassica has shown that specialised herbivores do not necessarily feed on seeds (Lucas-Barbosa et al. 2013)*.

We added additional information on this topic to the Introduction and Discussion. Please see also the answer to the editor’s comment above.

*Also, it should be outlined that the results found are quite specific for systems were pollinators are at the same time herbivores, and may not be broadly applicable to other pollination system such as bee pollination. This point should be made clear at some point in the paper*.

This point has now been addressed in the Discussion section.

*There are also a lot of errs and other problems in the manuscript as outlined below. The Result and Discussion section is broadly mixed with text belonging to the Materials and Methods section. The Introduction has a slightly arrogant tone at some places, sounding as if most previous studies investigating similar topics were inferior to this study, this is inappropriate given the fact that each study, including the one by the authors, has its strengths and weaknesses*.

*Introduction: “as with floral scent, the function of nectar on plant fitness has been studied in only a few plant systems”. Fitness effects of nectar has indeed been studied in many systems by adding nectar to rewardless flowers*.

This literature is now more respectfully acknowledged.

*Introduction: “again this single floral trait was studied only in the context of a single insect-plant interaction under glasshouse conditions”. Again, many of the studies that added nectar to flowers investigated the trait in the natural pollinator community context*.

Thank you, this point has now been corrected in the manuscript.

*Introduction: “attractive function of flowers”. This term is not appropriate here and at other places, because nectar is also a part of it. Use “floral signals” instead*.

Thank you. We changed this term accordingly.

*Results and Discussion: “these results demonstrate that native populations harbor nectar-free ‘cheating’ plants at low frequencies”. But the mechanism for the maintenance is not shown; the nectar-free individuals may just be spontaneous mutants with very low fitness. For a frequency dependency scenario, as outlined in the introduction, measurements of fitness in the nectarless individuals are necessary*.

This is an important argument and applies to any ecological study. These measurements however would require years of seedbank experiments, evaluating the viability of seeds in the soil after remaining in the ground for several years. This is something which is out of the scope of the current study. We added a sentence to the Methods section, mentioning the lack of data on the longevity of seeds in the seedbank of the plants used in this study.

*Results and Discussion: “large hawkmoths like* M. sexta *cover much larger distances each night and rely on direct perception rather an memory in visiting their food sources”. Moths also use learning when exploring new food sources, see e.g. (*[65]*)*.

This information, as well as the citation has been added to the manuscript.

*Results and Discussion: “It is intriguing that it is the reward rather than the attractant that influences the female hawkmoth’s oviposition decisions”. Reward is also an attractant, actually it is the primary attractant for pollinators*.

Figure 1
*is very confusing: there are three different axis labels, but only two different dots – where is nectar sugar concentration indicated? Normally, different axis labels in a scatter plot show the association of two traits in the same individuals – here it is seemingly two traits that are shown independently, thus it would be better to use two separate graphs. For the inset it says: “Mean (*+/-*SE) standing nectar volume in newly opened flowers at the end of the nectar production period (5-6am) in three native phenotypes (Ut-WT, 83, and 84) found within a separate sample of 424 native* N. attenuata *plants that were screened for the presence of nectar.” Now does the graph show nectar amounts of three individuals? Why are there error bars for one individual? Does the statistical test compare individuals? For nectar measurements, standardise to amounts per individual flower*.

We thank the reviewer for pointing out a mislabeling in the original figure, which is likely the cause of the confusion. “Nectar volume” should have been labeled “nectar sugar”, and a new corrected version of the figure is now included in the manuscript. The inset shows the mean nectar volume per flower of 7-9 plants of each of three genotypes, as described in the figure legend and the Method sections. The distinction between attractants and rewards is now more clearly articulated in the revised Introduction. Very few flowers openly display their nectar reward, so that it can function as a visual attractant. While many nectars are scented, it remains unclear how important this scent is in attracting flower visitors in comparison to the usually much larger amounts of scent released from other parts of the flower. Hence for most flowers, the attractants and the rewards are uncoupled.

*For BA emission, give absolute amounts in ng/flower, not peak area. Using peak area you may reach wrong conclusion regarding the differences in emission (Results and Discussion, first paragraph), as dose response curves in a mass selective detector for any compound may not be linear. It also makes the scent amounts non-comparable to other studies, because different MSD detectors may have different dose/response ratios*.

Done.

Reviewer #2:

*I really enjoyed reading this excellent study. The results were quite interesting and have important implications for those working on plant-insect interactions. All of my comments are on presentation and say nothing about the methodological accomplishments of this work and the data presented*.

*The Introduction could use some reworking. The first paragraph has a weak transition into floral scent and introduces florivores too early (“Floral scent plays a central role in attracting both insect pollinators and florivores to flowers”), and awkwardly brings in species differences (“Single traits (e.g. single floral volatile compounds) are thought to influence individual pollinating species differently”). In these first two paragraphs you discuss scent and nectar independently, but avoid discussing studies that have manipulated the two traits together and theory underlying why the two should synergize (or not), which is far more relevant to your work. The paragraph starting “The influence of nectar quantities on herbivory…” – the focus/purpose of this section is unclear. It introduces cheating, but in an awkward way and seems to cover a lot of unrelated ground. This should/could be tightened up. I'm unclear why rare nectar-free plants would trick pollinators, but not herbivores as well. Some of these ideas could be fleshed out much better in this paragraph*.

We completely agree that the Introduction needed recrafting and this has now been accomplished largely guided by these excellent suggestions. We are unaware of a literature of studies that examine fitness consequences of experimental manipulations of scent and nectar independently and simultaneously, but would be happy to do so if the reviewer could give us a pointer to this literature. We greatly appreciate the suggestion of motivating the work with Bell’s 1986 ESS model!

Your important question about why rare nectar-free plants would trick pollinators, but not herbivores as well, is a good one, and one that we can answer with a good bit of authority as we know much about what it takes to be an herbivore on this plant, particularly for *M. sexta*. The pollinator function of the plant’s interaction with this species is mediated by two relatively simple traits, BA and nectar, and the relationship is based on changing the apparency of the flower and providing a reward for a particular behavior. With regard to the plant’s interaction with the herbivore, the plant has a 5 layered suite of defense/tolerance/avoidance responses that are all elicited by *M. sexta* larvae elicitors, which all kick into gear when the egg hatches to produce a larva. The defense responses include indirect defenses, which like the pollinator interaction involve changing the apparency of the plant, albeit to predators of the herbivores. Previous work (Allmann and Baldwin 2010) demonstrated that some of the VOCs that mediate these indirect defenses also influence *Manduca*’s oviposition decisions. We have not added this point to the text as it would take a couple of paragraphs (and many citations) to flesh out this idea in a cogent manner, but we would be happy to do so if you think that it’s an important point to address. It would require a rather extensive set of citations, as the details of the 5 layers of defense/avoidance/tolerance responses are elaborated in about 250 publications.

*Your discussion of biological effects relative to your statistics seems overly loose. In several cases you make strong statements about effects that are not supported by your stats. For example: “for the more specialized pollinator,* M. sexta*, floral scent attraction is more important than the nectar reward”. But on*
Figure 2*, both of those treatment bars have a ‘b’ over them meaning that the two treatments are the same. Another example: “It is intriguing that it is the reward rather than the attractant that influences the female hawkmoth's oviposition decisions”. This is not true – on*
Figure 2*, both reward and attractant had strong effects on oviposition. In most of these cases you're referring to magnitude differences, some of which are not statistically supported, so you need to tighten up your language throughout when discussing biological effects relative to what your figures/stats show*.

Thanks for these helpful comments. All statements not supported by statistics have been vetted from the manuscript.

*Another related statistical term issue, you misuse the term ‘additive’ in a few spots (“Again, we found an additive negative effect in plants unable to produce both traits” and “In contrast to the pollination results, we found no additive effect in the CHALxSWEET9 cross”)*. *Most people would refer to additivity in terms of whether the effect of treatment A+B is the sum of the individual effects from A and B tested separately. I don't think this is how you're using it. Unfortunately you can't really test for additivity very well in many of these figures because your individual manipulations of scent and nectar have such strong effects on seeds/eggs that the expected additive outcome would a negative number!*

We removed this term from the manuscript.

*One last statistical comment – it might be easier to analyze your data as a two-way ANOVA rather than a one-way because then you can statistically disentangle the main effects of scent and nectar and their interaction effect, since this is a fully crossed design*.

A two-way ANOVA would not be appropriate, as the requirements for such an analysis are not fulfilled. In most cases the variances within the different lines are not comparable, and as a consequence the data is not homoscedastic, and a *Friedman* signed rank test had to be used.

You might add more consideration in the Discussion to your manipulations relative to nature. Although you establish the existence of nectar-less flowers, there were no scent free flowers in the wild so your manipulation for that trait is a theoretical assessment, which is different, or biologically could inform the consequences of plants lowering floral emission.

As BA emission is temporary and genetically variable there are times in the night when one plant is emitting BA, while the neighboring plant is not. We added a paragraph to the Results section describing this variability.

*Can you comment on the physiological costs of your manipulation to plants? You focus your study on the ecological costs of flowers in terms of herbivore attraction, but people will also be interested in the direct costs, i.e., Do your manipulated lines show any evidence of enhanced growth, flower production, etc. when volatiles/nectar are silenced? These traits are presumed to be quite costly so their silencing must benefits plants in some way*.

There are no obvious physiological costs associated to the production of BA or nectar. Flowers of all four lines produced equal amounts of seeds from hand pollinations (statistics are given in the Methods section). If nectar or scent production represented a physiological drag for the plant, we would expect early flowering from the rewardless plants, but the entire phenology of the plants is completely comparable across all four lines. We added a sentence to the Methods section describing this. The only thing we did not test was the production of flowers over the entire life time of a plant. This is something we have to address in future studies. In the experiments presented in this manuscript, plants were always reduced to the same number of flowers, so that all replicates would have the same visual “floral apparency”.

*This is not a criticism of how you performed the study – I like the anther removal – but I'd be very curious to see your study repeated without the removal. Do you have any idea how your data would look on normal plants? In other words*, *can tobacco selfing entirely negate the fitness costs of reduced pollinator visits you observed in*
Figure 2*?*

*N. attenuata* can produce just as much seeds from selfing, as it can from outcrossing. The reference for this observation is now added to the Results section: “*N. attenuata* is capable of producing the full set of seeds of comparable mass and number from self-pollination as from outcrossing (4).” The main difference is the advantage of having more genetically variable seeds, when being outcrossed. We address this question in a small paragraph in the Results section now.

*End of Results and Discussion: “We conclude that both pollinators, as well as herbivores shape the evolution of floral traits and that it makes little sense to study floral traits as if they only mediate pollination services”. I don't disagree with your conclusion here, I'm just wondering if you can make a broader case for the importance of floral traits for herbivores here or elsewhere. Part of me feels like the* Manduca *system is unique because the pollinator and herbivore are one and the same like in* Yucca*. I know this is not necessarily true, but I feel like you can make a stronger case for the relevance of this to many (most?) plant-insect systems for those who might believe it's an oddity to the* Manduca*-tobacco interaction*.

We have added a paragraph discussing the relevance of our results to other plant-insect systems at the end of the Discussion section.

Reviewer #3:

*[…] These experiments are not only elegant, but they are very difficult to accomplish as a choreography (germinate the plants, fly across the planet, make sure the RNAi worked, make sure they don't die, get the timing right for the natural visitors). Buried in the Methods are references to setbacks and lost field seasons, which underscore the degree of difficulty behind these assays. I have several questions about methods and presentation which can easily be dealt with by the authors. My other points are primarily requests that they modify some of their statements about prior studies on floral cheating. Again, I think these are easily dealt with, but I would like to see less dismissiveness in the authors' tone – empty flowers are a very old idea in pollination biology, and there is a large body of theoretical and experimental work on floral enemies shaping floral ecology and evolution. The beauty of this study, as in previous studies by these authors, is in its surgical manipulation of the traits in question and the inclusion of several floral visitors, some detrimental, in the design, with real fitness consequences as the response variables*.

Thanks for these insights as well as the appreciation of what’s involved in making this work happen. We have reworked the treatment of the literature and hopefully the tone of the literature review is now one of appreciation of being able to stand on the shoulders of others to see farther.